# Azeotropic Distillation-Induced Self-Assembly of Mesostructured Spherical Nanoparticles as Drug Cargos for Controlled Release of Curcumin

**DOI:** 10.3390/ph15030275

**Published:** 2022-02-23

**Authors:** Long Chen, Xin Fu, Mei Lin, Xingmao Jiang

**Affiliations:** 1Key Laboratory for Green Chemical Process of Ministry of Education, School of Chemical Engineering & Pharmacy, Wuhan Institute of Technology, Wuhan 430205, China; chenlong0729@126.com; 2Nanjing Zhongwei Biomaterials Research Institute Co., Ltd., Nanjing 210008, China; tiger6105180@163.com; 3Institute of Clinical Medicine, Taizhou People’s Hospital Affiliated to Nantong University, Taizhou 225300, China; l_mei@163.com

**Keywords:** drug delivery, curcumin, azeotropic distillation, self-assembly, hydrophobic, mesoporous silica nanospheres

## Abstract

Methods of large-scale controllable production of uniform monodispersed spherical nanoparticles have been one of the research directions of scientists in recent years. In this paper, we report an azeotropic distillation-induced evaporation self-assembly method as a universal method, and monodispersed hydrophobic ordered mesoporous silica nanospheres (MHSs) were successfully synthesized by this method, using triethoxymethylsilane (MTES) as the silica precursor and hexadecyl trimethyl ammonium bromide (CTAB) as the template. SEM and TEM images showed good monodispersity, sphericity, and uniform diameter. Meanwhile, SAXS and N_2_ adsorption–desorption measurements demonstrated a highly ordered lamellar mesostructure with a large pore volume. The model drug, curcumin was successfully encapsulated in MHSs for drug delivery testing, and their adsorption capacity was 3.45 mg g^−1^, which greatly improved the stability of curcumin. The release time when net release rate of curcumin reached 50% was extended to 6 days.

## 1. Introduction

In recent decades, drug delivery controlled by carrier systems has been demonstrated to have successful applications in the diagnosis and treatment of various diseases [1,2,3]. Mesostructured spherical nanoparticles are promising intracellular delivery systems for anticancer, immunomodulatory drugs and cell activity modulators, etc. [4,5]. The cellular uptake of nanoparticles by living cells is strongly size-dependent [6]. Small nanoparticle size (≈50 nm) is most efficient for the intracellular delivery [1,7]. The development of a suitable nanostructured carrier system with good biocompatibility and selective delivery of drugs to target cells is the central problem of nanomedicine. Curcumin is a natural bioactive substance, which has been of great interest to researchers due to its wide range of biological activities and pleiotropic therapeutic potential such as antioxidant, anti-inflammatory [8,9,10,11,12], antibacterial, antifungal, antiviral, antiprotozoal, and antiparasitic activities [11,13,14,15], but its application has been strictly limited because of its poor solubility in water, short half-life, low bioavailability, and pharmacokinetic profile.

Compared to general organic carriers such as liposomes [16], micelles [17], PLGA [18], cyclodextrin [19], viruses, etc., mesoporous silica nanoparticles have the significant advantage of sustained release profile [20,21], good biocompatibility, and large drug loading capacity, which is largely depending on their tunable surface chemistry and particle size, uniform pore size, high surface area. Therefore, mesoporous silica nanoparticles have attracted great research attention as cargos for delivery and controlled release of various drugs. Tang and co-workers found that mesoporous silica nanospheres modified by hydrophobic groups showed enhanced hydrothermal stability and delayed release of hydrophobic drug [22]. Highly monodispersed mesoporous silica nanospheres are needed to control the delivery rate [23]. While the method of large-scale controllable production of uniform monodispersed spherical nanoparticles has been one of the research directions of scientists in recent years. Brinker and his co-workers succeeded in fabricating mesoporous silica nanospheres by developed aerosol-assisted evaporation-induced self-assembly (EISA) progress [24]. The monodispersed spherical silica nanoparticles can be obtained after a few seconds of EISA progress, which greatly improves the production efficiency. Various templates (CTAB, Brij-56, Brij-58, P123) have been used to control the pore size and mesostructures of silica [25]. Then, they synthesized protocells by fusion of lipid bilayers to the mesoporous silica nanospheres, and this structure was prominent for drug delivery [1,3,5,26,27,28,29]. However, mesoporous silica nanospheres made using a commercial atomizer (Model 3076, TSI, Inc., St Paul, MN, USA) have a wide size distribution (~50–1000 nm), requiring separation. Although the electrospray aerosol generator (Model 3480, TSI, Inc., St Paul, MN, USA) can produce high concentrations of monodisperse submicron particles with diameters ranging from 2 nm to 100 nm [30], its yield is too low (Liquid flow rate: 50 to 100 nL min^−1^). Large-scale controllable production of MHSs with uniform particle size by aerosol-induced EISA is still a challenge.

Here, we report a facile synthesis method based on azeotropic distillation-induced self-assembly to prepare MHSs with good monodispersity and controllable uniform size. This method is easy to operate, and can solve the deficiencies in the synthesis of MHSs by aerosol-assisted EISA, such as wide size distribution, small specific surface area and pore volume. In addition, it can be used for the synthesis of many functional nanomaterials, and can be effectively applied to large-scale industrial production. In this report, the water, ethanol, benzene and CTAB form a stable reverse microemulsion under stirring and MTES has enough time to be hydrolyzed after adding to the system. The rising temperature causes the solvent to evaporate as an azeotrope, then the water phase and benzene will be condensed, the water phase can be separated by a water separator, and the benzene can reflux back to the reaction system. Each MHS can be formed from one single aqueous droplet in the microemulsion after evaporation and removal of water. The usual aerosol-assisted EISA is not able to synthesis of hydrophobic ordered porous silica with MTES, due to the fast evaporation and hydrolysis rate which cannot match with slow self-assembly rate of MTES. As a result, the relatively slow evaporation rate of azeotropic distillation is necessary for continuous self-assembly of MTES into highly ordered porous structures.

## 2. Results

### 2.1. Characterization of MHS Samples

Field emission scanning electron microscopy (FESEM) and transmission electron microscopy (TEM) images of the washed and dried MHS samples are shown in Figure 1. The MHS samples are uniform in size and spherical in shape (Figure 1a,b,d), and their particle size could be adjusted by the amount of CTAB/MTES mole ratio (Table 1). The particle size of MHS-1 was about 30 nm, while the particle size of MHS-2 is about 132 nm, because of the CTAB/MTES mole ratio of MHS-2 is lower. From the HRTEM image of MHS-1 (Figure 1c), CTAB produces particles exhibiting a highly ordered lamellar mesostructure.

To determine the pore ordering of the synthesized MHS samples, Small Angle X-ray Scattering (SAXS) analysis was performed from 0.8° to 12° (Figure 2). MHS-1 and MHS-2 showed sharp peaks at a low angle (2θ = 3.41°) and two weak peaks at higher angles (2 θ= 6.81° and 10.18°) corresponding to the (100), (200), and (300) planes, respectively. The SAXS curve of MHS-1 shows that the washed and dried particles have an ordered lamellar structure. Compared to MHS-1, due to the decrease of CTAB/MTES mole ratio, it is difficult for silica-surfactant liquid-crystalline mesophase to grow in an orderly fashion in the process of self-assembly induced by azeotropic distillation, which leads the structural order of MHS-2 to decrease.

Figure 3 shows the N_2_ adsorption–desorption isotherms and the pore size distribution curve for washed and dried MHS-1 and MHS-2 samples. The samples were outgassed at 250 °C for 10 h before measurements. The MHS-1 (Figure 3a) exhibited a typical type Ⅳ isotherm with a H4 hysteresis loop, giving a large pore volume (1.208 cm^3^ g^−1^) and a narrow pore size distribution (centered at 4.3110 nm) (Table 1). In Figure 3b, MHS-2 showed a type Ⅲ isotherm with a H3 hysteresis loop, indicating that the pore size is non-uniform, and compared with MHS-1, its pore volume (0.654 cm^3^ g^−1^) and the surface area (257.452 m^2^ g^−1^) drastically decreased (Table 1). This was attributed to the fact that the CTAB/MTES mole ratio decreased, and the structural order of MHS-2 decreased. This is consistent with the results of SAXS analysis. All results mentioned above further demonstrate that MHS-1 with a smaller size, uniform pore size and higher specific area is more suitable for hydrophobic drug storage and release.

The surface properties of washed MHS-1 and MHS-2 samples with different heat treatment temperatures were measured by the contact angle test in Figure 4. Tablets of MHSs were prepared using a cylindrical stainless-steel die with a diameter of 1 cm. A pressure of 15 bar was applied for 10 min using a manual hydraulic press. The contact angle results of washed MHS-1 and MHS-2 dried at 60 °C were 121.2° and 120.5° (Figure 4a,c), which showed that the silica has good hydrophobic property. The contact angle of the particles increased to 126.3° and 129.9° (Figure 4b,d) after increasing drying temperature to 300 °C, indicating that the free water and hydroxyl groups on the surface of MHS samples decreased with the increase of temperature. The thermal, chemical and physical properties of MHS-1 and MHS-2 were measured at different atmosphere by simultaneous thermogravimetry and differential scanning calorimetry (TG-DSC) in N_2_ (Figure 5a,c) or air (Figure 5b,d) at a constant heating rate of 10 °C min^−1^ in the temperature range between 30 °C and 800 °C. Compared with the data under N_2_ conditions, one strong exothermic peak appeared at 420 °C under air conditions, suggesting oxidation of methyl groups by oxygen. This result demonstrated a high thermal stability for MHSs. The mesopores were covered by lipophilic-CH_3_ groups, enabling curcumin molecules to easily enter and stay in the pores.

### 2.2. Adsorption and Release Experiment of Curcumin

After loading with curcumin, MHS-1 and MHS-2 were accordingly marked as MHSAC-1 and MHSAC-2, respectively. Figure 6 showed the FTIR spectra of MHS-1, MHSAC-1, MHS-2, MHSAC-2 and curcumin samples. The FTIR peaks in the ranges of 1026–1125 cm^−1^, 772–801 cm^−1^ and 434–441 cm^−1^ correspond to the asymmetric, symmetric stretching and bending modes of the Si-O-Si [31], respectively (I and III in Figure 6). A sharp peak appearing at 2974 cm^−1^ can be assigned to -CH_3_ group and an absorption peak at 1275 cm^−1^ belongs to Si–CH_3_ stretching vibrations [31]. The FTIR peaks at 3468 cm^−1^ belong to OH groups of Si-OH on the surface of the MHS samples. In the FTIR spectra for curcumin (Ⅴ in Figure 6), a sharp peak at 3511 cm^−1^ is assigned to the phenolic O-H stretching with a broad band at a range from 3100–3400 cm^−1^, which is due to the -OH group (in enol form). The strong peak at 1627 cm^−1^ is associated with mixed C=O and C=C species of curcumin. Another strong band at 1603 cm^−1^ is attributed to the symmetric aromatic ring stretching vibrations C=C ring. The 1509 cm^−1^ peak is assigned to the C=O, and the C-O-C stretching peak of ether at 1027 cm^−1^ [32]. After curcumin was adsorbed, a new absorption band belonging to the heptadiene-dione chromophore group of curcumin appeared in the range of 1429–1627 cm^−1^ (II and IV in Figure 6). Other FTIR peaks belonging to MHS samples had no obvious shift. These results suggested that curcumin molecule had been adsorbed to hydrophobic silica [33]. It was also found that the absorption band intensity at 1429–1627 cm^−1^ of MHSAC-1 sample was much stronger than that of MHSAC-2 sample as a result of more curcumin loaded in MHSAC-1. All these results demonstrate that curcumin was successfully encapsulated in the as-synthesized MHS-1 samples.

Curcumin was encapsulated in hydrophobic mesopores by repeated heating and cooling of curcumin solution (V_water_:V_ethanol_ = 1:1) and MHS samples [34]. The remaining curcumin solutions after removal of MHSAC-1 and MHSAC-2, were accordingly marked as C-1 and C-2, respectively. Figure 7a shows the UV-vis spectra of 3.94 mg L^−1^ curcumin solution, C-1 and C-2. A sharp absorption peak of curcumin in a mixed solution of water and ethanol (V_water_:V_ethanol_ = 1:1) appeared at 432 nm. A series of curcumin solutions with different concentration were prepared, and their absorbance at 432 nm was measured to obtain the standard curve of curcumin solution (Figure 7b). After curcumin was adsorbed, the intensity of the absorption peaks for C-1 and C-2 decreased as a result of attachment of curcumin to the MHS-1 and MHS-2. C-1 exhibited a much lower absorption peak than C-2, indicating that higher adsorption capacity of MHS-1 due to larger specific area and pore volume under the same conditions. After calculation, the curcumin adsorption capacity of MHS-1 was 3.45 mg g^−1^, while it was only 0.91 mg g^−1^ for MHS-2. It was proved that a mesoporous carrier with a high specific surface area, large pore volume, and appropriate pore size (larger than the kinetic diameter of the drug) would be beneficial for improving the adsorption capacity.

As shown in Figure 8a, the in vitro sustained release process of curcumin was investigated by means of UV-vis spectroscopy. Curcumin was released from MHSAC-1 for 21 days in phosphate-buffered saline (PBS, water, pH = 7.4). Figure 8b shows the in vitro release kinetics of curcumin from MHSAC-1 samples in PBS (water, pH = 7.4), which was calculated according to the standard curve of curcumin in Figure 7b. As shown, 70.6% of absorbed curcumin released slowly from MHSAC-1 samples in PBS, which lasted for 21 days. On about the sixth day, the net release rate of curcumin just reached 50% and then then rate of release became slow. It was important to extend the release time for practical controlled release. These results proved that the MHS nanospheres with a small size, larger specific area and pore volume (MHSAC-1) displayed a sustained release of curcumin and have great application potential in the study on the controllable slow release of hydrophobic drugs.

## 3. Discussion

Usually, because the atomization process is uncontrollable, the diameter distribution of aerosol droplets obtained by commercial atomizers (such as Model 3076, TSI, Inc., St Paul, MN, USA) is wide. Therefore, it is difficult to achieve uniformity of as-synthesized particle size. Some scholars have added a microwave radiation zone at the back end of commercial atomizers to cause aerosol droplets to break up due to overheating under microwave radiation. Due to the surface tension of droplets, it will make the aerosol droplets in the carrier gas maintain the maximum total contact area and minimize the Gibbs free energy, so that the particle size of the droplets entering the drying zone tends to be uniform.

The self-assembly induced by azeotropic distillation is designed based on this principle. On the one hand, it has some similarities with aerosol-assisted self-assembly, beginning with a homogeneous solution of soluble silica and surfactant prepared in ethanol/water solvent with c_0_ << critical micelle concentration (cmc). CTAB as a stabilizer [35] can disperse the water phase into small droplets with uniform size in the oil phase. Additionally, due to the surface tension of the droplets, this stirred and heated system tends to maintain the maximum total contact area, which can cause the Gibbs free energy to be the lowest. The hydrolysis of MTES in the droplets is controlled, which limits the excessive growth of silica particles. With the slow separation of the water phase in the azeotrope and the reflux of benzene, the evaporation of the solvent creates a radial gradient of surfactant concentration from surface of each droplet to inside that steepens in time [36]. As the surfactant on the droplet surface first reaches the critical micelle concentration (cmc) [24], the ordered silica-surfactant liquid-crystalline mesophase grows radially inward from the surface. Finally, as the solvent continues to decrease, the silica-surfactant mesophase dries and shrinks to a sphere. For all the nanoparticles that can be synthesized by aerosol-assisted self-assembly, the self-assembly induced by azeotropic distillation can be applied, and the uniformity of particles can be ensured.

On the other hand, it is also different from aerosol-assisted self-assembly. Azeotropic distillation can adjust the evaporation rate of azeotrope by controlling the temperature of the system. In a typical synthesis, it takes about 0.5 h to get 0.5 mL of condensed azeotrope, and all ethanol and water in the system can be separated for as long as several hours (It only takes a few seconds for ethanol and water to evaporate in the process of aerosol-assisted self-assembly). This ensures that there is sufficient time for ordered self-assembly of MTES and CTAB liquid crystal mesophase. Therefore, this method can also be used to synthesize some nanoparticles with special morphology, such as cube shape [37] and rod shape. Of course, the self-assembly process induced by azeotropic distillation also has some shortcomings. The system is an inverse microemulsion system which depends on stirring to achieve uniform dispersion. Stirring speeds that are too violent or too slow may lead to irregular morphology of the final particles.

The MHSs prepared by self-assembly induced by azeotropic distillation show a good ability to control the sustained release of hydrophobic drugs. The release time at which the net release rate of curcumin reached 50% was extended to 6 days, which was much slower than curcumin-conjugated silica nanoparticles (7 h) [33], L- methionine encapsulated by hollow mesoporous silica nanoparticles (50 min) [34], and curcumin loaded by mesoporous silica nanoparticles (functionalized by 3-aminopropyltriethoxyorthosilane) (50 h) [38]. These results prove that this material can also be expected to be used in the encapsulation of other fat-soluble drugs, such as taxol, methotrexate, doxorubicin, etc.

## 4. Materials and Methods

### 4.1. Reagents and Instruments

All reagent-grade chemicals were used as received without further purification, and ultra nanopure distilled water (18.25 MΩ·cm) was used in all experiments. Curcumin, triethoxymethylsilane (MTES) and hexadecyl trimethyl ammonium bromide (CTAB) were purchased from Shanghai Aladdin Biochemical Technology Co., Ltd. (Shanghai, China). Ethanol, isopropyl alcohol, and benzene were bought from Sinopharm Chemical Reagent Co., Ltd. (Shanghai, China).

X-ray powder diffraction (XRD) patterns were recorded on a Rigaku Ultima IV powder diffractometer with a Cu Kα radiation source (λ = 1.5406 Å, 40 kV, 100 mA). Field emission scanning electron microscopy (FESEM, SIGMA Zeiss, Germany) and transmission electron microscopy (TEM, FEI Tecnai 30, 300 kV, Philips) were applied for characterization of the morphology of the samples. Fourier transform infrared spectroscopy (FTIR) were measured with a Tensor-Ⅱ spectrometer (Bruker Co., Germany) by averaging 64 scans with a spatial resolution of 4 cm^−1^. UV-vis absorption spectra were measured by Shimadzu UV-3600 UV-Vis-NIR spectrophotometer. The contact angles were measured by the XG-CAMA static contact angle tester. Thermal behavior of the samples was analyzed by thermogravimetry and differential scanning calorimetry (TG/DSC) (NETZSCH STA 409 PC, Germany). Labsys Evo simultaneous thermal analyzer was used to test the thermal stability of mesoporous hydrophobic silica. BET-surface area was measured by N_2_ adsorption–desorption at liquid nitrogen temperature using an Autosorb-iQ2-MP (Quantachrome) gas sorption system. Specific surface areas were calculated using the Brunauer–Emmett–Teller (BET) model, and the pore size distributions were evaluated from the adsorption branches of the nitrogen isotherms using the Barrett–Joyner–Halenda (BJH) model.

### 4.2. Synthesis of MHS Samples

MHS samples were synthesized by azeotropic distillation-induced self-assembly, as defined in Figure 1. In a typical synthesis, MHS-1 was prepared as follows: 1 g CTAB was dissolved in a solution of 7.5 mL deionized water, 18.5 mL ethanol and 74 mL benzene. The mixture was poured into a 250 mL three-necked flask mounted with a Dean-Stark trap followed by stirring at room temperature for 30 min to form a reverse microemulsion. Then 0.2 mL MTES was added into the above solution, which was kept at 45 °C for 2 h. With the hydrolysis of MTES molecules and formation of silanols, silica species became more hydrophilic and enriched in the aqueous droplets as a result of phase equilibrium. With constant hydrolysis of MTES and the enhanced hydrophilic of silica precursors, silica species continuously diffused from the benzene phase into water–ethanol droplets due to decreased solubility. Evaporation of water and ethanol from the droplets by azeotropic distillation at 64.9 °C (azeotrope composition: ethanol 18.5%, benzene 74% and water 7.5%) increased the concentrations of the silica species and CTAB, which led to self-assembly of the micelles into liquid crystalline mesophase. Meanwhile, hydrophilic inorganic precursors were also condensed into ordered porous silica with the liquid crystal as templates. The solution was then heated to 115 °C to remove the solvent. Subsequently, the samples were cooled to room temperature, collected and washed with a solution (deionized water and isopropyl alcohol, the volume ratio of 1: 1) to remove the CTAB, and further dried at 60 °C for 6 h. The preparation process for MHS-2 is the same as for MHS-1 except for adding 3 mL MTES instead of 0.2 mL MTES.

### 4.3. Adsorption and Release Experiment of Curcumin

MHSAC-1 was prepared as follows: 0.1 g MHS-1 was dissolved in 100 mL 3.94 mg L^−1^ curcumin solution (V_water_:V_ethanol_ = 1:1), followed by repeated heating and cooling several times until the color of the solution did not change. The samples were collected by centrifugation at 13000 rpm for 10 min and dried at 60 °C for 1 h. The preparation process for MHSAC-2 is the same as for MHSAC-1. The in vitro release kinetics of curcumin from MHSAC was as follows: Sixteen MHSAC samples of 0.01 g each were dissolved in 10 mL phosphate-buffered saline (PBS, water, pH 7.4). Then these samples were stirred at 100 rpm with a magnetic stirrer at 37 °C. After a period of time, the solution was centrifuged at 7000 rpm for 3 min to separate curcumin from the bottom of phosphate-buffered saline. The separated curcumin was dissolved in 10 mL mixed solution (V_water_:V_ethanol_ = 1:1, pH 7.0) then the UV-vis absorption spectra were recorded.

## 5. Conclusions

In summary, ordered mesoporous hydrophobic silica nanoparticles (MHSs) with a uniform size were successfully one-step synthesized by an azeotropic distillation-assisted method with MTES as precursor and CTAB as an ordered mesoporous template. The obtained MHSs exhibited high monodispersity, good sphericity, and large pore volume, with a highly ordered lamellar mesostructure, while the particle size can also be adjusted. This method solved the deficiencies in the synthesis of MHSs by aerosol-assisted EISA, such as wide size distribution, and small specific surface area and pore volume. Curcumin was successfully encapsulated in MHSs, and their adsorption capacity was 3.45 mg g^−1^, greatly improving the stability of curcumin. The release time after which net release rate of curcumin reached 50% was extended to 6 days. Curcumin can be released slowly from MHSs, guaranteeing that curcumin has enough time to reach and inhibit cancer cells, bacteria, fungus, etc. MHSs also have great application potential in the study on the encapsulation of other hydrophobic drugs for drug delivery such as taxol, methotrexate, doxorubicin, etc.

## 6. Patents

There was a patent (Patent number: CN104876230B) resulting from the work reported in this manuscript and it was licensed.

## Data Availability

Data is contained within the article.

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
