# Peer review of "Azeotropic Distillation-Induced Self-Assembly of Mesostructured Spherical Nanoparticles as Drug Cargos for Controlled Release of Curcumin"

_pharmaceuticals, 2022, doi:10.3390/ph15030275_

Round 1

Reviewer 1 Report

This manuscript describes the development of ordered mesoporous silica nanoparticles using azeotropic distillation assisted method and their use as curcumin delivery system.

In my opinion, this manuscript lacks of novelty since there are many works in the literature describing studies on the curcumin-loaded silica nanoparticles. Thus, the novelty of this work should be explicitly explained.

Additional comments are as followed.

  1. The N2 adsorption-desorption isotherms mainly of MHS-1 sample are not correct as any adsorption/desorption hysteresis loop was not observed which is characteristic for mesoporous materials. Please repeat these experiments and correct the Figure 3 and the results of the table 1. Before these measurements, did you outgas the samples? At what temperature and for how many time? Please add this info to the experimental part?
  2. Please give experimental details for the Contact angle measurements. What substrate did you use, how did you apply the samples on the substrate, etc.?
  3. CA as well as TGA/DSC results for the MHS-2 are missing. Please add them.
  4. The DSC diagram in Figure 5a is not correct. Please do it again.
  5. It is not clear for me how the authors concluded from FTIR results that the encapsulation of curcumin to the MHSs took place through the enolic hydroxyl groups. The cumarin FTIR spectrum should be added and compared with the spectra of MHSAC-1/2 and MHSs. The assignment of FTIR spectra should be clearly rewritten, explaining all the changes and the appearance of the new peaks. The FTIR analysis should be presented after the results of encapsulation and release experiments.
  6. The experimental part regarding the adsorption and release experiment of curcumin should be presented in details. As it is not clear if the process that was taken place was adsorption or absorption it would be better to characterize it as sorption or simply encapsulation. Additionally, how did the authors remove the non-encapsulated cumarin? How did you calculate the amount of the encapsulated cumarin and the absorbance capacity? In lines 280-281, the authors mentioned that they performed centrifuge to separate curcumin from PBS. Did they use the supernatant to calculate the amount of cumarin that encapsulated to MHS or cumarin received as a pellet and then dissolved in water/ethanol solution? Why did they use a mixture of water/ethanol to dissolve cumarin since it is water soluble at these concentrations? In line 275, they said that MHS was dissolved in 100 mL 3.95 mg 274 L-1 curcumin solution. Is this solution aqueous (only water) or a mixture of water/ethanol?
  7. Figure 7. Are the C-1 and C-2 solutions aqueous or water/ethanol solutions? What is the peak at about 360nm observed in the UV spectra of both C-1 and C-2? How did you measure the concentration of cumarin from the spectrum of C-1 since the peak at ~360nm almost overlapped with the peak at 425 nm.
  8. The cumarin release profiles should be described in more details. The authors only mentioned that after 21 days, 70.6 % of absorbed curcumin released slowly from MHSAC-1 samples in PBS.
  9. In the discussion part, the encapsulation and the release results should be discussed in comparison with the analogous results from the literature.
  10. Please correct some typo-errors.

Author Response

Dear reviewer,

Thank you for your comments and suggestions. We are sorry to reply you so late. Our modifications to the article are as followed.

  1. The abstract and introduction have been re-written, some important references supporting the statements have been added.
  2. The N2 adsorption-desorption experiments, contact angle measurements, FTIR, UV-vis and TGA/DSC results of MHS-1 and MHS-2 have been repeated, some new figures have been uploaded. The new outgassing procedure is 10 h at 250 ℃. The CA measurements were taken as followed: Tablets of MHSs were prepared using a cylindrical stainless steel die of 1 cm in diameter. A pressure of 15 bar was applied for 10 min using a manual hydraulic press. The magnification of SEM and FTIR range have been given.
  3. The experimental part regarding adsorption and release experiment of curcumin has been re-written in details, and the release kinetics (figure 8b) of curcumin compared with a recent research (Curcumin Conjugated Silica Nanoparticles for Improving Bioavailability and Its Anticancer Applications https://doi.org/10.1021/jf402894x).
  4. The standard curve of curcumin (figure 7b) has been uploaded to calculate the vitro release kinetics (figure 8b) of curcumin.
  5. Because the capacity of curcumin in MHS has not reached the maximum loading capacity, which is not comparable to some recent studies, some data have been deleted.
  6. Some figures image quality has been improved, the size of 3 and Fig 8 has been increased. Some English grammar and spelling mistakes have been corrected.
  7. Please see the attachment to get the details.

Thank you very much for your time and consideration.

Sincerely yours,

 Long Chen

Reviewer 2 Report

Dear Authors,  

The Manuscript ID: pharmaceuticals-1556279 is interesting and it could be published after a minor revision. I made myself some suggestions/corrections in the attached manuscript. Authors must pay attention to the red highlighted words/sentences.

Briefly, I suggested the following:

  • In Introduction, the authors must highlight the novelty/originality of their work.
  • English language and style must be improved.
  • Line 40: Replace “depend” with “depending”.
  • Lines 53, 143, 195, 202: Insert space. 
  • Lines 205-208: Attention to the format!  
  • Line 63: Please explain the use of the word: “delaminated”. 
  • All abbreviations must be detailed when they first appear in the text!
  • Lines 65, 220: I think the correct abbreviation is MHS, not HMS!
  • Lines 85-87: In the Table 1, The correct abbreviation is:

SABET

https://www.sciencedirect.com/topics/engineering/brunauer-emmett-teller-method

because “BSA” is the abbreviation of the protein: Bovine Serum Albumin.

  • Line 113: delete “different”.
  • Line 114: Reformulate!
  • The section: “2. Adsorption and release experiment of curcumin” must be carefully revised as follows:
    • Lines 135, 148: Replace “absorption” with “FTIR”.
    • Lines 135-137: Please insert a reference regarding the attribution of Si-O-Si band:

Barbinta-Patrascu, M. E., Ungureanu, C., Badea, N., Constantin, M., Purcar, V., Ispas, A., Bioperformances of honey-phytonanosilver in silica materials, Journal of Optoelectronics and Advanced Materials 22(5-6), 310-315, 2020; WOS:000563834000017 (https://joam.inoe.ro/articles/bioperformances-of-honey-phytonanosilver-in-silica-materials/fulltext)

    • Line 141: The statement: “new peak in the range of 1429-1627 cm-1 appeared” is valid only for the sample MHSAC-1. Please check and revise it!
    • Lines 141-142: The strong peak at 1627 cm-1 is observed only in the FTIR spectrum of the sample MHSAC-1. Authors must revise the sentence from the Lines 141-142.
    • Lines 152-153: In Figure 6, for FTIR bands visualization, please insert a zoom of the wavenumber range: 1700-1400 cm-1.
    • Line 157: Why did the authors not use the amount of 3.95 mg/L of curcumin that was used in the preparation of the samples MHSAC-1 and MHSAC-2 (see Figure 7 and the section: 4.3. Adsorption and release experiment of curcumin)?
    • Lines 163-164: Authors should give the mathematical expression for the calculation of adsorption capacity (mg/g) of curcumin.
    • Line 170: Authors should give the mathematical expression for the calculation of %release of curcumin.
  • Lines 211, 252, 282: Please replace “ml” with “mL”.
  • Line 282: Please complete the sentence by adding: “then the UV-Vis absorption spectra were recorded.”
  • Lines 273-282: Authors should give some mathematical expressions for the calculation of adsorption capacity (mg/g) and of %release of curcumin.
  • Lines 223, 292, 294: Delete “and”.

Line 369: Replace “small” with “Small”.

Author Response

Dear reviewer,

Thank you for your comments and suggestions. We are sorry to reply you so late. Our modifications to the article are as followed.

  1. The abstract and introduction have been re-written, some important references supporting the statements have been added.
  2. The N2 adsorption-desorption experiments, contact angle measurements, FTIR, UV-vis and TGA/DSC results of MHS-1 and MHS-2 have been repeated, some new figures have been uploaded. The new outgassing procedure is 10 h at 250 ℃. The CA measurements were taken as followed: Tablets of MHSs were prepared using a cylindrical stainless steel die of 1 cm in diameter. A pressure of 15 bar was applied for 10 min using a manual hydraulic press. The magnification of SEM and FTIR range have been given.
  3. The standard curve of curcumin (figure 7b) has been uploaded to calculate the vitro release kinetics (figure 8b) of curcumin.
  4. Because the capacity of curcumin in MHS has not reached the maximum loading capacity, which is not comparable to some recent studies, some data have been deleted.
  5. Some figures image quality has been improved, the size of 3 and Fig 8 has been increased. Some English grammar and spelling mistakes have been corrected.
  6. Please see the attachment to get the details.

Thank you very much for your time and consideration.

Sincerely yours,

 Long Chen

Reviewer 3 Report

The article is technically sound and covers the scope of journal "Pharmaceuticals".

The article can be published after addressing below given points;

1- Abstract should be re-written, authors are advised to seek help of any scientific writer or experienced colleague.

2- Introduction is lacking references to support the statements and please write reasoning at some points which are directly related to current research.

such as Page 1 line 28,  Small nanoparticle size (< 50 28 nm) is preferred for the intracellular delivery [reference missing], Also please address why this size is preffered keeping current study in consideration.
Authors are advised carefully check and address similar issues within the whole manuscript to avoid delay in processing.

3- Please divide long sentences of introduction into short segments for readers ease.

4- Overall Figures image quality should be improved (important).

5-  Fig. 6(c) should be re-measured/corrected on FTIR keeping all parameters same or Authors should justify the noise peaks appearing in the respective graph.

6- Fig. 3 and Fig 8, please increase the size of graph to make it readable.

7- Please provide all important outcomes in the form of significant values directing towards current research.

8- Page 4, line 113, The surface properties of MHS-1 samples with different different heat treatment 113 temperatures were measured by the contact angle test in Figure 4. Different is mentioned twice, overall language is fine but authors are advised to check minor typing errors.

9- Materials and Methods
Please give details about magnification is SEM,
FTIR range of measurement
CA testing procedure or add respective references https://doi.org/10.3390/polym13081245.

Best of Luck!

Author Response

Dear reviewer:

Thank you for your comments and suggestions. We are sorry to reply you so late. Our modifications to the article are as followed.

  1. The abstract and introduction have been re-written, some important references supporting the statements have been added.
  2. The N2 adsorption-desorption experiments, contact angle measurements, FTIR, UV-vis and TGA/DSC results of MHS-1 and MHS-2 have been repeated, the new outgassing procedure is 10 h at 250 ℃. The CA measurements were taken as followed: Tablets of MHSs were prepared using a cylindrical stainless steel die of 1 cm in diameter. A pressure of 15 bar was applied for 10 min using a manual hydraulic press. The magnification of SEM and FTIR range have been given.
  3. Some figures image quality has been improved, the size of Fig 3 and Fig 8 has been increased. Some English grammar and spelling mistakes have been corrected.
  4. Please see the attachment to get the details.

Thank you very much for your time and consideration.

Sincerely yours,

 Long Chen

Round 2

Reviewer 1 Report

The manuscript has been improved as most of the comments have been addressed. However there are some points that need improvement.  Specifically:

  1. The novelty of this work should be clearly presented in the last paragraph of the introduction as well as in the conclusions of the manuscript.
  2. The authors said that they added some new references but unfortunately I cannot find any.
  3. As these nanoparticles were proposed to be used as cumarin delivery system. The loading capacity of the nanoparticles as well as the drug efficiency should be presented.
  4. As I have asked in my previous report, in the discussion part, the encapsulation and the release results should be discussed in comparison with the analogous results from the literature.
  5. Please correct some typo-errors.

Author Response

Dear reviewer,

Thank you for your comments and suggestions again.

  1. As you said, we will clearly point out the novelty of this work in the last paragraph of the introduction and in the conclusion of the manuscript.
  2. In line 36,163,165 and 222, we added some new references, and in line 222, the release kinetics (figure 8b) of curcumin was discussed in comparison with a recent research (Curcumin Conjugated Silica Nanoparticles for Improving Bioavailability and Its Anticancer Applications https://doi.org/10.1021/jf402894x). We will also put more comparisons with the analogous results from the literature in the discussion part.
  3. We will continue to check and correct typo-errors in the article.

The new revised version will be uploaded soon.

Thank you very much for your time and consideration.

Sincerely yours,

 Long Chen

Round 3

Reviewer 1 Report

The authors were taken under consideration almost all my suggestions and now I think that the manuscript is suitable for publication.

A minor correction: As these nanoparticles were proposed to be used as cumarin delivery system, their absorbance capacity should be mentioned in the abstract and the conclusions.

Author Response

Dear reviewer,

Thank you for your suggestions again. The adsorption capacity of the material has been added in the abstract and the conclusion.

The new revised version has been uploaded.

Thank you very much for your time and consideration.

Sincerely yours,

 Long Chen